# Multidrug Resistance (MDR) and Collateral Sensitivity in Bacteria, with Special Attention to Genetic and Evolutionary Aspects and to the Perspectives of Antimicrobial Peptides—A Review

**DOI:** 10.3390/pathogens9070522

**Published:** 2020-06-29

**Authors:** András Fodor, Birhan Addisie Abate, Péter Deák, László Fodor, Ervin Gyenge, Michael G. Klein, Zsuzsanna Koncz, Josephat Muvevi, László Ötvös, Gyöngyi Székely, Dávid Vozik, László Makrai

**Affiliations:** 1Department of Genetics, University of Szeged, H-6726 Szeged, Hungary; deakp@brc.hu; 2Ethiopian Biotechnology Institute, Agricultural Biotechnology Directorate, Addis Ababa 5954, Ethiopia; birhanaddisie@gmail.com; 3Institute of Biochemistry, Biological Research Centre, H-6726 Szeged, Hungary; 4Department of Microbiology and Infectious Diseases, University of Veterinary Medicine, P.O. Box 22, H-1581 Budapest, Hungary; Fodor.Laszlo@univet.hu; 5Hungarian Department of Biology and Ecology, Faculty of Biology and Geology, Babeș-Bolyai University, 5-7 Clinicilor St., 400006 Cluj-Napoca, Romania; gyenge_ervin@yahoo.com (E.G.); gyongyi.szekely@ubbcluj.ro (G.S.); 6Institute for Research-Development-Innovation in Applied Natural Sciences, Babeș-Bolyai University, 30 Fântânele St., 400294 Cluj-Napoca, Romania; 7Department of Entomology, The Ohio State University, 1680 Madison Ave., Wooster, OH 44691, USA; klein.10@osu.edu; 8Max-Planck Institut für Pflanzenzüchtungsforschung, Carl-von-Linné-Weg 10, D-50829 Köln, Germany; zskoncz@mpipz.mpg.de; 9National Cereals and Produce Board, Mombasa 80100, Kenya; jmuvevi@gmail.com; 10OLPE, LLC, Audubon, PA 19403-1965, USA; lotvos@comcast.net; 11Institute of Medical Microbiology, Semmelweis University, H-1085 Budapest, Hungary; 12Arrevus, Inc., Raleigh, NC 27612, USA; 13Centre for Systems Biology, Biodiversity and Bioresources, Babeș-Bolyai University, 5-7 Clinicilor St., 400006 Cluj-Napoca, Romania; 14Research Institute on Bioengineering, Membrane Technology and Energetics, Faculty of Engineering, University of Veszprem, H-8200 Veszprém, Hungary; vozik.david@drv.hu or or

**Keywords:** MDR, intrinsic/acquired resistance, collateral sensitivity, negative frequency-dependent selection, experimental evolution, pangenome, global dissemination, mobility patterns of resistance genes, adaptive evolution

## Abstract

Antibiotic poly-resistance (multidrug-, extreme-, and pan-drug resistance) is controlled by adaptive evolution. Darwinian and Lamarckian interpretations of resistance evolution are discussed. Arguments for, and against, pessimistic forecasts on a fatal “post-antibiotic era” are evaluated. In commensal niches, the appearance of a new antibiotic resistance often reduces fitness, but compensatory mutations may counteract this tendency. The appearance of new antibiotic resistance is frequently accompanied by a collateral sensitivity to other resistances. Organisms with an expanding open pan-genome, such as *Acinetobacter baumannii, Pseudomonas aeruginosa*, and *Klebsiella pneumoniae*, can withstand an increased number of resistances by exploiting their evolutionary plasticity and disseminating clonally or poly-clonally. Multidrug-resistant pathogen clones can become predominant under antibiotic stress conditions but, under the influence of negative frequency-dependent selection, are prevented from rising to dominance in a population in a commensal niche. Antimicrobial peptides have a great potential to combat multidrug resistance, since antibiotic-resistant bacteria have shown a high frequency of collateral sensitivity to antimicrobial peptides. In addition, the mobility patterns of antibiotic resistance, and antimicrobial peptide resistance, genes are completely different. The integron trade in commensal niches is fortunately limited by the species-specificity of resistance genes. Hence, we theorize that the suggested post-antibiotic era has not yet come, and indeed might never come.

## 1. Introduction

The rapid and ongoing spread of antibiotic resistance poses a serious threat to global public health [1]. Considering the complexity of the sophisticated resistance mechanisms [2,3], recruiting cooperating scientists with different backgrounds seems reasonable and necessary. This review is addressed to committed scientists, biologists, geneticists, and evolutionary biologists who may be attracted by fundamental, rather than applied, research perspectives. We focused on selected aspects. One is the danger posed by globally spreading monoclonal pathogens (*Acinetobacter baumannii* [4,5,6], *Pseudomonas aeruginosa* [7,8,9]), with an expanding open pangenome, being aware that they are not the only ones displaying a threat.

Antibiotic resistance was discovered in 1940 [10]. The selective pressure of antimicrobials is favorable for resistant clones to survive and spread [11]. The emergence of antibiotic multiresistance (MDR) in pathogenic bacteria has become alarming in the last decades [12]. MDR appeared not only in human clinical pathogens [13,14,15,16] but also in zoonic bacteria [17,18], veterinary pathogens [19,20,21,22,23,24,25,26], and in plant pathogens [27,28,29,30,31,32]. Although application technologies are improving for plant pathogens [33], the trend has been that the use of antibiotics as plant medicines has gradually been restricted [34,35]. A precondition for elaborating appropriate therapies is having a better understanding of antibiotic resistance mechanisms [36] in Gram-negative [13,16,37] and Gram-positive [38,39,40,41] pathogens. New “bugs” are continuously appearing, and new antibiotics are needed. Conly and Johnson asked in 2005 “Where are the new antibiotics?” [42]. The tantalizing answer came 11 years later: Antibiotics are “right under our nose!” [43]. Unfortunately, they still have not made it to market [44], at least not in the required number [45,46].

## 2. Multidrug Resistance: Updated Terms and Definitions

The genomes of different bacterium species harbor silent genes that code for resistance, but in the absence of antibiotics it is not manifested. Consequently, a large pool of potential of antibiotic resistance genes is hidden in various niches. The term antibiotic resistome covers the collection of all antibiotic resistance genes, including those associated with both pathogenic and non-pathogenic bacteria, but not ones which produce antibiotics [47]. Pál and his associates provided a comprehensive characterization of resistance genes, mobile genetic elements (MGEs), and bacterial taxonomic compositions for 864 metagenomes from humans (n = 350), animals (n = 145), and external environments (n = 369), all deeply sequenced using Illumina technology. They concluded that antibiotic-polluted environments are probably under-estimated transmission routes, and indeed reservoirs for antibiotic resistance [48].

### 2.1. Antibiotic Resistance as a Phenotype

The resistance genotype [49,50,51] determines the resistance phenotype. Take the evolutionary history of *Mycobacterium tuberculosis* resistance to second-line anti-tuberculous as a scholarly medical genetic example to illustrate the simple Mendelian genotype–phenotype relations [52,53,54,55].

Genetic suppression of the cellular autolytic system of pneumococci causes simultaneous resistance to penicillin, D-cycloserine, and phosphonomycin [56], providing an example of the iso-allelic inheritance phenotype [57,58]. Physiological suppression is similar [56], providing an example of the old genetic term, phenocopy [59,60]. The clarification of genotypic/phenotypic relations is a prelude to the application of clinical metagenomics [61].

### 2.2. The Location and Harboring of the Resistance Genes

The resistance gene is a peptide-coding open reading frame, genetically regulated as structure gene of an operon. The higher level of organization is the antibiotics resistance cassette [62,63,64]. The respective resistance gene(s) could be localized either on the chromosome [51], or in a plasmid [65,66,67,68,69], either as non-conjugative [62], conjugative [70,71], or in an episome [72].

### 2.3. Insertion and Excision

The resistance gene as an integron component [71,73] can be harbored by a mobile genetic element capable of inserting into, and excising from, another DNA molecule like a plasmid. This has been reviewed by several authors [74,75,76,77,78,79].

### 2.4. Homolog Recombination

Some mobile elements are capable of recombining with the bacterial chromosome, allowing transfer of large segments, as shown with vancomycin and penicillin resistance transfer, at least in Gram-positive (*Enterococcus*) pathogens [80,81]. Horizontal gene transfer (HGT) could happen even between the different bacteria taxa [82,83].

### 2.5. Intrinsic Resistance (IR)

IR is the phenotypic expression of a resistance gene originally present, resulting in structural and functional changes of the original gene product [84,85]. This can be either a decomposing enzyme [86,87,88], or the target site of the drug [89,90]. The most efficient resistance mechanisms are the multidrug efflux pumps [91,92,93,94].

When resistance to antimicrobial compound is a phenotypic expression of a resistance-encoding gene coming from outside via horizontal gene transfer HGT [95,96,97,98,99,100], it is called “acquired” resistance [85,101]. The gene from outside had been harbored by a plasmid and was taken up from the environment, such as from soil [102,103] or the gastrointestinal [83,99] microbiota community. Expansion of KPC-producing *Klebsiella pneumoniae* with various mgrB mutations giving rise to colistin resistance [104], and the bla (VIM-1) metallo-beta-lactamase producing. *E. coli* was published to spreading over in a university hospital in Greece [105]. These are examples of concerted activities of mobile genetic elements, including insertion sequences, transposons, and gene cassettes/integrons, and has been summarized by Partridge et al. [79]. Horizontal gene transfer (HGT) could even happen between different bacteria taxa [82,83].

The novel phenomenon called distributive conjugal gene transfer (DCT) was recently discovered in mycobacteria. DCT involves the transfer of chromosomal DNA between mycobacteria and, most significantly, generates trans-conjugants with mosaic genomes of the parental strains [106].

### 2.6. Definitions of Antibiotic Polyresistant Strains

Antibiotic resistances [2,3] are classified as follows: multidrug resistant (MDR) is not susceptible to at least one representative from each of three categories of selected antimicrobial compound families [3]. Extreme drug-resistant (XDR) is not susceptible to at least a single representative of all but very few categories of antimicrobial compound families. Pan-drug resistant (PDR) is not susceptible to any of the tested representatives of all known antimicrobial compound families [3].

### 2.7. The Updated List of the “ESKAPE” Polyresistant Pathogenic Bacterium Species

In 2006, the Antimicrobial Availability Task Force (AATF) of the Infectious Diseases Society of America (IDSA) prepared a review that highlighted frequently resistant pathogens to licensed antimicrobials [36], characterized by high antibiotic resistance, and extremely versatile MDR phenotypes which are responsible for many nosocomial infections [36]. As for their clinical significance [107], the updated the list of six ESKAPE Pathogen Bacterium Species appeared in 2008 [18]. The six letters are the initials of the genera of these bacteria: *Enterococcus faecium*, *Staphylococcus aureus, Klebsiella pneumoniae, Acinetobacter baumannii, Pseudomonas aeruginosa,* and *Enterobacter* spp. [108,109,110,111,112]. There is “NO DRUG” against them and “NO ESKAPE” from them [109]. They include causative pathogens of nosocomial diseases, such as ventilator-associated pneumonia [113,114], infections in burn wounds [115], and pathogens which “escape” from antimicrobial agents [109,116,117]. Immune-compromised patients are most exposed to ESKAPE pathogens causing bacteremia [118].

The ESKAPE list has recently been updated by the World Health Organization (WHO) [119]. It includes the carbapenem-resistant *Klebsiella pneumoniae, Acinetobacter baumannii, Pseudomonas aeruginosa*, carbapenem-resistant and third-generation cephalosporin-resistant Enterobacteriaceae, clarithromycin-resistant *Helicobacter pylori*, fluoroquinolone resistant *Campylobacter* spp., *Neisseria gonorrhea*, and *Salmonella typhi* as community-acquired infection-causing pathogens. The highest-ranked Gram-positive members are vancomycin-resistant *Enterococcus fecalis* and *Ent. faecium*, as well as the methicillin-resistant (MRSA) *S. aureus* [119]. The resistance mechanisms used by the ESCAPE pathogens were recently reviewed [120].

Some other dangerous Gram-negatives such as *Escherichia coli, Francisella tularensis*, and Gram-positives *Mycoplasma bovis and Bacillus anthracis* may be considered as putative “ESKAPE Club-Members” in the future. We discuss this option in Appendix A.

## 3. Resistance Problems Related to Gram-Negative Pathogens

The logical chain of events leading to multidrug resistance in Gram-negative pathogens is illustrated in Figure 1. **(LEFT).**

### 3.1. Β-Lactams vs. ESBL Resistance in Enterobacteriaceae and Klebsiella pneumoniae

The β-lactams were introduced to overcome penicillin resistance by preventing re-assembly of peptidoglycan bonds, which eventually leads to cell lysis [121,122]. Extended-Spectrum Β-Lactamase (ESBL) producing Enterobacteriaceae and *K. pneumoniae* pathogens are resistant to β-lactam-based antibiotics. The emergence of, and challenges by, ESBL-producing Enterobacteriaceae has been reviewed over the last three decades [123,124,125,126]. Pitout suggested that in vitro resistance to ceftazidime, and/or aztreonam, should be used as phenotypic markers of ESBL [127]. In Gram-negative bacteria, the β-lactamase production is frequently associated with reduced permeability of the outer membrane efflux [128].

Biochemistry has increased the number of β-lactamase enzymes with enlarged substrate specificities [129]. The substrate range includes cephalosporins (cefotaxime, ceftriaxone [130,131,132]), monobactam aztreonam [133,134,135], amino-penicillin combinations [136], lactamase inhibitor ampicillin-sulbactam [137,138,139,140], ureido-penicillins [141,142] including piperacillin/tazobactam [143,144,145,146], temocillin [147], piperacillin/tazobactam [148], and ceftolozane-tazobactam [149]. Furthermore, the appearance of new enzymes is not associated with the loss of ability to hydrolyze the earlier lactams, such as ampicillin [36]. In addition, the prevalence of ESBL production among *E. coli* and *Klebsiella* species is variable [13]. ESBL can be considered as first strike back from the Gram-negative pathogens or as the first “Wedge” from Nature.

### 3.2. Carbapenems: A Strong Antibiotic (“High Card”) to Beat the “Wedge” ESBL

To overcome ESBL problems, a new beta-lactam class of antibiotics, the carbapenems [150,151], was developed [152], including blood and respiratory isolates of *P. aeruginosa* [153]. As Breilh and his associates summarized, they act either as “slow substrate” β-lactamase inhibitors or by binding to penicillin-binding proteins [154]. This “value-added feature” of inhibiting β-lactamases serves as a major rationale for the expansion of this class of β-lactams [155]. Carbapenem antibiotics, including imipenem, are Gram-negative cell-wall synthesis interrupting molecules, which bind to penicillin-binding proteins [156]. Cilastatin is a human enzyme, dehydropeptidase, in the kidney that inhibits imipenem degradation and meropenem [156], as well as ertapenem [157]. Each one alone, or in combination, had been a putative last line of defense against multidrug-resistant Gram-negative organisms.

### 3.3. Carbapenem Resistance: The Second Strike Back from the Gram-Negative Pathogens (“Wedge” from Nature)

Since 2006, the number of carbapenem-resistant Enterobacteriaceae (CRE) has significantly increased [158,159,160]. A novel, epidemic, serine class-A type enzyme (KPC) is behind carbapenem resistance. It is encoded by the Bla (Oxa) gene family [161,162,163,164,165]. KPC exhibits powerful activity against all types of Beta-lactam molecules [166,167,168].

### 3.4. NDM1

A novel type of plasmid-encoding carbapenem-resistant metallo-β-lactamasecarbapenemase NDM 1 [169] was identified in 2008 in two Enterobacteriaceae isolates, both recovered from a Swedish patient transferred from India [170]. The emergence of NDM-expressing enterobacteria [171] and *Klebsiella* [172,173] was then reported from all continents, but it recently reappeared in Italy (Toscana, Z. Koncz, personal communication). There are publications on NDM 1 expressing *Acinetobacter* [174], *Pseudomonas* [175], and only one *E. coli* publication [105]. It has never been found in Gram-positive bacteria. *Klebsiella pneumoniae, Acinetobacter,* and *Pseudomonas* are spread by clonal dissemination [176], while *E. coli* and other Enterobacteriaceae spread through polyclonal dissemination [177]. Many carbapenem-resistance genes localized on mobile genetic elements have been previously reviewed [178,179,180,181,182,183,184].

### 3.5. “Dropped and Rediscovered” Colistin, as a Large Spectral Antibiotic (a “Trump Card”)

The cationic lipopeptides polymyxin B, E, and colistin [185] disrupt outer membranes (OMs) and have been used since 1959 for treating infections caused by Gram-negative MDR pathogens. However, nephrotoxic side effects were discovered [186]. Since the appearance of several new infections caused by MDR Gram-negative organisms [187,188], and intensive searches of old antibiotic options, colistin was rediscovered and considered to be a potential trump card [189,190,191,192,193,194].

### 3.6. Colistin Resistance: The Third Unexpected Attack (“Wedge”) from Nature

The first colistin-resistant mutant was reported in 1981 [195]. The appearance of colistin resistance, especially plasmid-mediated transferable ones [196,197,198,199,200,201], questioned whether the polymyxins should be considered as the last “life-savers” [202]. Colistin resistance is a consequence of post-translational modification, or loss, of the lipopolysaccharide (LPS) molecules [203]. The first colistin-resistant mutant was reported in 1981. The appearance of colistin resistance, especially the plasmid-mediated transferable ones, questioned whether the polymyxins were still be considered as the last “life-savers”. The mechanisms of acquired and intrinsic resistance of polymyxin in different bacteria have recently been reviewed by many authors [204,205,206,207,208,209]. The genetic factors behind colistin resistance are *mcr* genes [198,210,211]. Colistin resistance has been evolving under clinical conditions [212,213] in *A. baumannii* [214,215,216], *P. aeruginosa* [217], *K. pneumoniae* [218,219], and *Enterobacter cloacae* [220]. The evolution of colistin resistance was more than a one step process, requiring mutation in at least five independent loci synergistically, creating the resistant phenotype [211,212,221,222].

### 3.7. Efforts to Overcome MDR Problems in Gram-Negative Pathogens in the Absence of Omnipotent (“Jolly Joker”) Antibiotics

The potentiation of β-lactam antibiotics and β-lactam with β-lactamase inhibitor combinations was effective when used against MDR and XDR *P. aeruginosa*, using non-ribosomal tobramycin–cyclam conjugates [223,224]. A detailed evaluation of publications reviewing arguments for and against combining colistin and carbapenems controlling infections caused by carbapenem-resistant Enterobacteriaceae (CRE), *K. pneumoniae* carbapenemase (KPC)-producing bacteria, and carbapenem-resistant *A. baumannii* (CRAB), based on randomized controlled trials in which treatment with colistin was in combination with meropenem or rifampin, showed it did not work as expected. This demonstrates that the use of some polymyxin has always been necessary when either CRAB, CRE, or CRPA harboring metallo-beta-lactamases are at stake [225]. The combination strategy is especially useful when the applictiin design is syncronized with the results by obtained the recently published cassette assay [226] (aiming efficiently quantifying the outer membrane (OM) permeability of multiple β-lactams in carbapenem and colistin-resistant *Klebsiella. pneumoniae*) and enabling to rationally optimize the use of the dose of synergistic β-lactam antibiotics, [226].

## 4. Resistance Problems Related to Gram-Positive Pathogens

The logical chain of events leading to multidrug resistance in Gram-positive pathogens is illustrated in Figure 1. **(RIGHT)**.

*Streptococcus* and *Enterococcus* species are intrinsically resistant to beta-lactams [227]. They use beta-lactamases and/or just [228] penicillin-binding polypeptides [229]. This has been reviewed by Funda and his associates [230].

### 4.1. Methicillin-Resistant Staphylococcus aureus (MRSA)

Since the early sixties [231], multiresistant pathogen strains [232,233], including those resistant to vancomycin, have appeared [234]. MRSA is the causative agent in many diseases worldwide [235]. This includes nosocomial infections like pneumonia [236], spondylodiscitis [237], colonization of burns and wounds [238,239], endocarditis [240], and renal problems [241], all of which have greatly increased hospital death tolls [242]. Methicillin-resistant *S. aureus* (MRSA), and *S. epidermis* isolates, from companion animals have been reviewed [243,244]. Previously, the clinical use of vancomycin, synergistically combined with antibiotics, was suggested against poly-resistant clinical isolates [245]. However, vancomycin and daptomycin resistance have developed in the same patient within hours [246].

#### 4.1.1. The Molecular Basis of MRSA

Extreme methicillin resistance is polygenicly inherited [247]. The responsible chromosomal DNA segment of ~50 kb, named mec, consists of the open-reading frame mecA coding for the penicillin-binding protein 2a (PBP 2a), linked to the regulatory genes mecI and mecR1, which control mecA expression. A variable number of resistance determinants are also included [248]. Strains lacking mecA may show low-level methicillin resistance, due to modifications in native PBPs, or the expression of beta-lactamase [249]. However, there is only one publication on methicillin decomposing an enzyme [250], which has not been confirmed.

#### 4.1.2. The Accelerated Evolution of MRSA

MRSA evolution started when the mecA was acquired, probably when the penicillinase-resistant oxazolidines were introduced [251]. The genetic profile, antibiotic resistance and bacteriophage profiles, and the pulse-field patterns of contemporary epidemic MRSA clones were very similar to those of the early methicillin-sensitive (MSSA) isolates [252].

The evolutionary history of MRSA was reviewed by Antignac and Tomasz [253]. The donor *S*. *sciuri* strain has a mecA-homolog gene, capbpD, coding for the penicillin-binding protein PBP 4. The mecA gene is harbored by a SCCmec element and was horizontally transferred to an originally sensitive (MSSA) strain of *S. aureus* [253]. Since this discovery, an international working group (IWG-SCC) has been concentrating on the classification of “Staphylococcal Cassette Chromosome Elements” [254,255]. Another team (Cepheid Healthcare-Associated Infection (HAI) Consortium) has continuously been addressing the changing epidemiology of MRSA isolates, with the recognition of both “empty cassette” strains, where mecA is lost from the SCCmec cassette, and the emergence of SCCmec variants [256]. The first MRSA isolates, followed by applied Bayesian phylogenetic reconstruction, provided an option for reconstructing further details of the evolutionary history of the archetypal MRSA [257]. Harkins and his associates assumed the approximate date at which the earliest MRSA lineage harboring the SCCmec appeared was about the mid-1940s, the era of methicillin [257]. The research field of staphylococcal cell wall structure has gradually gained in its significance [258]. *Streptococcus pneumoniae* has a complex cell wall that plays a key role in contributing to pneumococcal resistance to lysozymes [259].

### 4.2. Enterococci: The Gram-Positive “Vanguards” of the “MDR Movement”

*Enterococcus* species *(Ent. faecium, Ent. faecalis, Ent. gallinarum*, and *Ent. cecorum*; for taxonomy, see Wikipedia) are facultative anaerobes that exist as commensals in the gastrointestinal tract of a variety of organisms, including humans [260]. Six penicillin-binding proteins (PBPs) were identified in the first clinical isolates [261]. From samples studied, *Ent. faecium* was found to be the most resistant to, and showed the lowest affinities for, penicillin, while *Ent. bovis* was the most penicillin-sensitive and showed the highest affinity [261,262,263]. When streptomycin was discovered [262], it was applied together with penicillin [262], and the synergistic effect seemed to work [264]. However, this combination provided a selective condition for MDR enterococci [263].

MDR enterococci strains are adapted to the gastrointestinal tract and can become the dominant flora [265,266]. The first epidemic MDR *Ent. faecium* strain emerged from animal and commensal strains [267,268]. Enterococci have been, and remained, a prominent Gram-positive pathogen in the SENTRY (Antimicrobial Surveillance) Program from 1997 until now [269]. The most common *Enterococcus* pathogen species are *Ent. faecalis* (64.7%) and *Ent. faecium* (29.0%) [269]. *Enterococcus faecum* became the most important nosocomial pathogen, posing a growing clinical challenge because of its rapidly evolving antibiotic resistance repertoire, practically against all clinically used antimicrobials. Enterococci use many spectacular genetic strategies [270,271,272].

The MDR enterococci have individual combinations of genuine antibiotic resistance mechanisms [38,39,40,41,273,274,275,276,277,278]. The resistance mechanisms include modification of drug targets [38], inactivation of therapeutic agents [272,279,280], and overexpression of efflux pumps [38,281,282]. For more details, see Appendix A.

Their dynamic cell envelopes serve as their first line of defense against antimicrobials [264,283]. Peptidognt, ycan (PG), is a well-established target for antibiotics [284,285,286,287]. Components, like teichoic acids, capsular polysaccharides (CPS), surface proteins, and phospholipids, can undergo modifications and reduce the susceptibility to antibiotics [288,289,290].

#### 4.2.1. Comparative Genomics of Enterococci: Species, Strains, Clades Resistance Groups

The recent way to identify of *Enterococcus* species has been based on a simple assay of the groEL gene [291]. Comparative genomics showed that the MDR *Ent. faecium* genetic clade A1 [291,292,293,294,295] had separated evolutionarily from the animal-adapted *Ent. faecium* [295,296,297] at about the same time as penicillin and streptomycin were jointly introduced into clinical use [297].

#### 4.2.2. Clinically Adapted and Non-Clinical *Enterococcus* strains

Clinically adapted *Enterococcus* strains are still evolving from commensal strains, such as CC17 from *Ent. foecium,* but not from *Ent. faecalis* [266,267,298,299]. Clinical and non-clinical strains of *Ent. faecium* have distinct structural and functional genomic features [300,301].

#### 4.2.3. MDR Potential of Enterococci

For the genetics and genomics of ampicillin resistance in *Ent. faecium*, see the comprehensive review by Zhang et al. [302]. Evolutionarily separated clinical strains intensively collect mobile genetic elements, resulting in alterations in hyper-mutability that lends *Ent. faecium* its remarkable genome plasticity [303]. Enterococci serve as donors of antibiotic resistance gene clusters, including vancomycin resistance [304,305,306,307,308,309] and daptomycin resistance [310], to other Gram-positive pathogenic micro-organisms including MRSA [310].

#### 4.2.4. Vancomycin-resistant *Enterococcus faecium* VRE [298] as a Leading Cause of MDR Hospital Infections

In the *Enterococcus* genus, the vancomycin resistance (VRE) gene clusters are classified into nine types according to their gene sequences and the organization (reviewed by Chen and Xu) [311]. The latest review on the genomics of *Ent. faecium* VRE summarized the new insights into VRE evolution, drug resistance, and hospital adaptation based on whole-genome sequencing [312,313].

Two genetically definable *Ent. faecium* populations are the hospital-adapted MDR (including vancomycin-resistant) isolates (Clade A) [268,293,312,313] and vancomycin-susceptible commensal strains (Clade B). The majority of five VRE isolates collected from different sites belong to VanA [269,314]. However, the isolate VanN [315], which had previously been found in chickens in Japan [316], and then was also found in a hospitalized patient in Canada [317], emerged recently from a commensal (CladB) isolate [268].

#### 4.2.5. *Enterococcus cecorum*: An Example for the Recent Speedy Development of Antibiotic Multiresistance in Genus *Enterococcus*

A normal commensal intestinal inhabitant is increasingly responsible for outbreaks of arthritis and osteomyelitis in chickens. Since 2002, *Ent. cecorum* has been recognized as the causative pathogen of enterococcal spondylitis (ES) [260,318,319,320,321,322,323,324,325,326,327,328]. *Enterococcus cecorum* was known as a harmless commensal of the gastrointestinal tract of chickens. However, in the last one-and-a-half decades, new pathogenic isolates of *Ent. cecorum* have been an increasingly significant cause of morbidity and mortality in broiler chickens, and frequent outbreaks are reported, although an environmental reservoir for pathogenic *Ent. cecorum* has not been found. Classical genetic analyses of *Ent. cecorum* demonstrated that strains with increased pathogenicity are genetically related, and share several virulence genes and antibiotic resistance genes, when compared to commensal strains. These pathogenic strains can be recovered from retail meat, and they may serve as a reservoir for further spread of antimicrobial resistance among other *Enterococcus* spp. [329]. For more details, see Appendix A.

#### 4.2.6. Vancomycin, Vancomycin Resistance; Daptomycin, Daptomycin Resistance

Vancomycin is a non-ribosomal (NRP)-templated glycopeptide from *Amycolatopsis orientalis* [330,331]. It was used against Gram-positive targets, both intravenously and per os [332,333]. When taken by mouth it is poorly absorbed. Vancomycin acts by inhibiting proper cell wall synthesis in Gram-positive bacteria [38,334].

Vancomycin resistance: After the discovery and introduction of the omnipotent antibiotic vancomycin, it was considered the last resort, or “card” in the “card game” of pharmaceutical science, for Gram-positive pathogens [38]. Since the appearance of vancomycin-resistant enterococci strains in the late 1980s, the number of resistances has been steadily rising, both in Enterococci [14,271,335], and Staphylococci [336,337]. Many resistance isolates appeared, often with life-threatening consequences [337,338,339,340]. As an alternative to the generation of completely new substances, novel approaches have focused on structural modifications of vancomycin to overcome these resistances and to restore its efficacy against vancomycin-resistant enterococci [334,340]. The most recent knowledge about the mechanism of vancomycin resistance has been summarized [341].

#### 4.2.7. Daptomycin (DAP)

Daptomycin [342,343,344,345,346,347,348,349,350,351,352] is a cyclic anionic lipo-(dipepsi) lipopeptide, which includes three D-amino acid residues (D-asparagine, D-alanine, and D-serine) linked to a hydrocarbon tail, ten carbons in length, derived from decanoic acid [347]. It is biosynthesized by *S. roseosporus* from a soil sample gathered from Mount Ararat, Armenia [348]. DAP was introduced as a new drug against *S. aureus* (MRSA) and vancomycin-resistant enterococci (VRE), vancomycin-intermediate *S. aureus* (VISA), and penicillin-resistant *S. pneumoniae* [344,349,350,351].

A spectacular genetic analysis, including transposon mutagenesis [352] and molecular cloning [353], allowed discovery of the biosynthetic pathway of DAP. There are perspectives of improved production of DAP by exploiting the potential of genetic manipulation of secondary metabolite biosynthesis [354]. The total chemical synthesis has also been done [354].

#### 4.2.8. Daptomycin Resistance DAP (R)

The discovery of the mode of action of daptomycin [355] helps to handle daptomycin resistance problems. Intensive studies on daptomycin-resistant (DAP-R) mutants have also been discovered in enterococci [349] [356,357]. This has resulted in revealing the mechanisms of DAP-R in different Gram-positive bacteria [38,39,40,41,273], including *B. subtilis* [358], *Enterococci* [359,360,361,362], *Ent. faecalis* [361], *Ent. faecium* [362,363,364], and *S. aureus* [365,366,367,368]. Mutations of DNA “mismatch repair” genes in a DAP-R pleiotropic phenotype were discovered in a clinical isolate of *Ent. faecium* [369].

## 5. The Efforts to Discover Omnipotent (“Jolly Joker”) Antibiotics to Overcome MDR Problems

### 5.1. Teixobactin: The First Omnipotent (“Jolly Joker”) Antibiotic Active Against Gram-Positive Targets

The first antibiotic without detectable resistance was discovered in Kim Lewis’ laboratory. Teixobactin inhibits cell wall synthesis by binding to a highly conserved motif of precursor peptidoglycan lipid II and precursor of cell wall teichoic acid-lipid III (a) [370]. No *S. aureus* nor *Mycobacterium tuberculosis* mutants have been found to be resistant to teixobactin so far [371]. However, resistance mediated by D-stereospecific peptidases cannot be ruled out in the future [372]. Teixobactin is a non-ribosomal peptide (NRP) antibiotic, which contains D-amino acid. The total synthesis and structure–activity relationships of teixobactin have been conducted [373], and lactam and ring-expanded analogs are also available [374]. Teixobactin provides a new perspective for using antibiotics in the treatment of mycobacterial infections [375].

### 5.2. Narrow/Spectral Pluripotent Antimicrobial Peptides

We consider antimicrobial peptides (AMPs) as any polyamide (or even biopolymer with ester, thioester, or otherwise modified backbone) that can be made on a contemporary chemical peptide synthesizer, and that exerts antimicrobial activity [376]. The antibiotic-resistant bacteria show widespread collateral sensitivity to antimicrobial peptides [377], as discussed later in detail.

AMPs are either of natural origin, such as those isolated from insects [378], or designed [379] either by computer [380], following QASR-studies on the designer molecule [381] or after synthetic (experimental) precursor manipulation [382]. The natural AMPs are part of the inherent immune system in all known taxa but Archaea, [380,383], and many of them were found to be effective against multiresistant pathogens, usually, with high selectivity [383] Encodings and models for antimicrobial peptide classification for multi-resistant pathogens have then been developed, [383]. Based on conclusions drawn from QSAR and in vitro bioassays the putative promising chemical derivatives will then be structurally optimized [384]. Synthetic candidate derivatives including those substituted with unusual amino acids [385] share unique features such as serum stability, lack of side effects, and owing intracellular target specificity. The discovery of a new natural AMP molecule is usually followed by quantitative structure/activity (QSAR) studies [381], and then the design and optimization of analog molecules [379,380,381,382,383,384,385] then the best ones finally will chemically be synthesized. The most promising molecular family is the proline-arginine rich peptide (PrAMP) [386]. including pyrrochorrycins [386,387], apeadicins [388,389], and oncocins [390]. Some of them are in a preclinical stage [391,392]. All of these are narrow spectral AMPs. Some were more efficient in vivo than in vitro, supposedly due to cooperation with the innate immune system of the host to be protected [376,386,387,388,389,390,391,392].

### 5.3. Large Spectral Pluripotent (“Jolly Joker” Candidate) Non-Ribosomal Encoded (NRP) Biosynthetic Antimicrobial Peptides

Omadacycline aminomethylcycline is a semisynthetic derivative of tetracycline, and it is active against many Gram-negative bacteria and Gram-positives, including MRSA, *S. aureus*, *S. pneumoniae*, β-hemolytic streptococci, *Enterococcus* VRE, and Enterobacteriaceae in vitro [393] and in vivo in clinical practice [394]. Similar activity has also been found in the thermostable antimicrobial peptide discovered in Hungary in *Xenorhabdus budapestensis* and *X. szentirmaii* [31,395,396,397,398,399] and later identified as fabclavine [400,401,402,403,404,405].

### 5.4. Efflux Pump Inhibitors (EPIs)

Efflux pump inhibitors (EPIs) had great potential, but the putative drug candidates were toxic [406]. Two new carbapenem–β-Lactamase inhibitor combinations were introduced [151]. In the last years, several hundred thousand people developed multidrug-resistant tuberculosis (MDR-TB), requiring a new effective treatment. It was found that efflux pumps play an important role in the evolution of drug resistance. New strategies are required to mitigate the consequences of the activity of efflux pumps [130].

## 6. Adaptive Evolution: Trends and Mechanisms in Relation to MDR-Posed Danger

Antibiotic multiresistance has an obviously increasing worldwide tendency an evolutionary trend. The real question is whether the evolution of MDR is an irreversible one-way street.

### 6.1. Resistance vs. Persistence: Darwinian and Lamarckian Approaches

The proper approach for trying to slow down the rapid evolution of resistant bacteria requires a full understanding of the mechanisms of adaptive evolution [68,407,408,409]. The problem with the similarities between resistant and tolerant phenotypes is that tolerance can be associated with the failure of antibiotic treatments [410]. The correct and precise definitions of the terms of tolerance, persistence, persistent, and resistance have been provided in “opinion papers” [410] and confirmed by the Consensus Statement found in Balaban and her associates [411].

The phenotypic similarities between resistance and persistence led to Darwinian- and Neo-Lamarckian-inspired research approaches. Resistance is a phenotypic expression of a resistance allele and is specific for an antibiotic with a unique structure. This allows the microorganism to grow in the constant presence of the antibiotic within a concentration range [412].

This can be quantitatively characterized by a Minimal Inhibiting Concentration (MIC). On the other hand, tolerance (the synonym of persistence) [413] provides a general, not antibiotic-specific protection. Inheritable persistence is a pleiotropic phenotype with loss-of-function alleles from many genes with completely different functions [414], called toleromes by Brauner et al. [410].

Non-inherited persistence is inducible [415,416,417], fitness-decreasing [418], and in a dormant stage [413]. This is the phenotype of the epigenetic subpopulation [419,420,421] that remains in the culture after the actively respiring bacteria have been vanquished by antibiotics [422,423].

The Minimum Duration for killing (MDK99) is the quantitative parameter designed to evaluate the tolerance level of the population treated with antibiotics [410]. The epigenetic persisting subpopulation, with a non-inherited persistent phenotype, has a clear adaptive value [424,425,426,427]. It serves as a sheltered evolutionary reservoir [428] from which antibiotic-resistant mutants may emerge [426,427]. Potentially, “persistence invites resistance” [429], but it appears that persistence may, but not must, precede resistance [429,430,431]. Persistence-based antibiotic resistance research efforts are of key importance from the aspect of elaborating anti-persistence strategies [432,433,434,435,436,437,438,439,440,441], but this is out of the scope of this review.

### 6.2. An Introduction to Experimental Evolution

The 60-year history of genomic research, which helped reconstruct mobile genetic elements harboring genes responsible for multidrug resistance of pathogens, has recently been reviewed [407,442]. The spread of many multidrug-resistant (MDR) bacteria is predominantly clonal. Interestingly, international clones/sequence types (STs) of most pathogens emerged and disseminated during the last three decades [443]. This kind of evolutionary processes can be monitored or experimentally recapitulated. Bottlenecks reduce the size of the gene pool within populations of all, with implications for their subsequent survival. By reducing genetic diversity, bottlenecks may alter individual or population-wide adaptive potential [444]. Here we discuss some other aspects directed to antibiotic resistance.

#### Resistance Evolution and Mobile Genetic Elements in Enterococci

Gilmore and his associates wanted to determine how the enormous accretion of mobile elements affect the competitive growth of enterococci in the gastrointestinal tract consortium [303,445]. They observed that the prototype clinical isolate strain (V583) was actively killed by the gastrointestinal tract flora, while commensal enterococci flourished. They found that the death of V583 resulted from lethal crosstalk between accumulated mobile elements, and that this crosstalk was induced by a heptapeptide pheromone produced by native *Ent. faecalis* present in the fecal consortium [303]. They concluded the accumulation of mobile elements in hospital isolates of enterococci can include those that are incompatible with native flora. That is, in the absence of antibiotics, wild type (commensal) bacteria can win the competition with MDR pathogens.

### 6.3. Resistance Is Not a Positive Selection Marker in a Commensal Niche—The Lesson Learned from Population Genetics and Monitoring Evolutionary Processes

The virulent, globally disseminated, multidrug-resistant lineage ST131 of *E. coli* can cause urinary tract infections and bacteremia [443,444]. To differentiate ST131 from the larger *E. coli* population associated with a disease, Kallonen and his colleagues isolated samples from various parts of England for genomic analysis under the framework of a systematic 11-year hospital-based survey [446]. They concluded that antibiotic resistance must not have been the predominant reason for the prevalence of *E. coli* lineages in this population, but the frequency of *E. coli* lineages in an invasive disease was driven by negative frequency-dependent selection, acting in the commensal niche outside of the hospitals. This research team later showed that the predominant multidrug resistant *E. coli* clones, under the influence of negative frequency-dependent selection, are prevented from rising to dominance in a population [447].

In a long-term (2001–2014), worldwide analysis of enterococci and vancomycin-resistant enterococci (VRE) causing invasive infections, Mendes and his team evaluated the prevalence and in vitro susceptibility of enterococci and VRE in bloodstream infections in European and US hospitals [448]. Through a SENTRY Surveillance Program, Mendes and his associates found that the VRE rates among *Ent. faecium* increased both in Europe and the USA. The multidrug-resistant (MDR) phenotype was a positive selective trait of antibiotic stress under hospital conditions.

### 6.4. The Loading Capacity of the Bacterial Genome Is Limited—The Lesson Learned from Experimental Evolution

*Escherichia coli* is an ideal organism for experimental evolution. Most changes to its metabolic network that survived over the past 100 million years, which are adapted genetically to the environment, are due to horizontal gene transfer, with a very low contribution from horizontal gene transference [407,442]. Experimental microbial evolution studies use controlled laboratory populations to evaluate the mechanisms of evolution and are based on molecular, genomic, and maybe transcriptomic and proteogenomic analysis over several generations of evolving populations in controlled, pre-established conditions in a morbidostat [449,450,451] or in a DiVERGE [452]. The srecently developed technolog, enables researchers to carry out testing on empirical prediction ns of a given evolutionary theory [453,454]. The data from the literature gave an impression that a new antbitoc resistance is a positive selection marker in antibiotic stress conditions but is a negative selection marker in commensal milieu (as demonstrated on Figure 2A).

#### 6.4.1. Evolutionary Trends in The Presence of Antibiotics

Toprak and his associates experimentally recapitulated the evolution of antibiotic resistance developed in *E. coli* against several different antibiotics, and they found a gradual increase in resistance levels of each antibiotic tested during the experimental period [455]. The whole-genome sequencing of the evolved strains revealed mutations including resistant ones to the given drug or resistant ones to more than one drug. Resistance could be developed either in a stepwise manner in replicates (trimethoprim) or via diverse combinations of mutations in different genes (chloramphenicol and doxycycline). For interactions between the presence of an antibiotic and resistance evolution, the evolutionary dynamics were monitored by whole-genome deep sequencing every 3 to 4 days in another Gram-negative pathogen organism, *P. aeruginosa*. During the experimental period, the resistance level to colistin increased by one order of magnitude in 10 days and two orders of magnitude in 20 days, supposedly due to mutations in the *mut*S mutator gene [456].

##### Resistance, Fitness, and Compensatory Mutations

The antimicrobial pressure may drive the evolution of its resistance, which may be associated with reduced bacterial fitness. There is evidence that the probable “evolutionary price” of a new antibiotic resistance is lowering fitness, and it must be determined genetically [457,458,459,460]. The phenotypic consequences are reflected in the geometry adaptations [461,462]. However, compensatory mutations [461] may change the pattern and might be an explanation of the dissemination of colistin-resistant *A. baumannii* isolates [63,463]. This might be an explanation for the dissemination of colistin-resistant *A. baumannii* isolates [203]. Using precise, high-throughput fitness measurements for genome-wide *E. coli* gene deletion strains, with eight antibiotics, the width of the distribution of fitness effects (so-called DFE) mutations, which occur spontaneously from one antibiotic to another, is lower than in the absence of the antibiotic stress [462]. Unlike the DFE mutations, the magnitude of the changes in tolerated drug concentration, resulting from genome-wide mutations, are similar for most drugs but are exceptionally small [464].

##### Collateral Sensitivity

In a large-scale laboratory evolutionary experiment, Lázár and her associates discovered a novel trend in the evolution of antibiotic hypersensitivity in *E. coli* called collateral sensitivity [377,465,466,467,468]. Some populations which became adapted to amino-glycosides have especially low fitness in the presence of several other antibiotics. The whole-genome sequences of 63 independently evolved amino-glycoside-resistant strains demonstrated multiple mechanisms based on reduced proton-motive force (PMF) through the inner membrane. The comparison of captured determinants of the antibiotic cross-resistance interaction network demonstrated that convergent molecular evolution was predominant across antibiotic treatments, and the resistance encoding mutations simultaneously enhanced sensitivity to many other drugs [465]. This phenomenon is called collateral sensitivity [and illustrated on Figure 2B. As mentioned before, antibiotic-resistant bacteria show widespread collateral sensitivity to antimicrobial peptides [377].

##### Antibiotic and Antimicrobial Resistance Genes are of Different Mobility Patterns

The antibiotic-resistant bacteria not only showed an unexpectedly high frequency of collateral sensitivity to antimicrobial peptides, while cross-resistances between the strains were rather rare [390], but clinically relevant multidrug-resistance mutations also increased susceptibility to antimicrobial peptides. As for mechanisms, collateral sensitivity in multidrug-resistant bacteria arise partly through regulatory changes shaping the lipopolysaccharide composition of the bacterial outer membrane. This information allows the identification of antimicrobial peptide and antibiotic combinations that enhance antibiotic activity against multidrug-resistant bacteria and slow down de novo evolution of resistance [377,468]. Furthermore, while the antimicrobial peptide (AMP) resistance genes are widespread in the gut microbiome, their rate of horizontal transfer is lower than that of antibiotic resistance genes [469]. Gut microbiota culturing and functional metagenomics have revealed that AMP resistance genes originating from phylogenetically distant bacteria have only a limited potential to confer resistance in *E. coli*, an intrinsically susceptible species [469].

#### 6.4.2. Evolutionary Trends of Antibiotic Resistance in the Absence of Antibiotic Exposure

Antibiotic use is the main driver in the emergence of antibiotic resistance. In a set of experiments with *E. coli*, it was demonstrated that in the absence of an antibiotic and resistance evolution, drug-resistance frequently declined within 480 generations during exposure to an antibiotic-free environment.

The extent of resistance declination seemed antibiotic-specific and driven by mutations influencing both the resistance level and fitness probably pleiotropically [469]. It was concluded that a phenotypic reversion to the antibiotic-sensitive state may be mediated by the acquisition of additional mutations while maintaining the original resistance mutations [469]. This result might be an important argument against the selective virtue of antibiotic resistance in general.

However, in a set of experiments on another Gram-negative pathogen, *P. aeruginosa*, an opposite trend was demonstrated. It is difficult to interpret this contradiction. It was demonstrated that selection in the absence of antibiotics did co-select for decreased susceptibility to several antibiotics. This was considered as experimental evidence confirming the selective virtue of antibiotic resistance in a commensal niche [468]. The interpretation of these results by the authors is that resistance evolved coincidentally in response to other selective pressures. Thus, genetic adaptation of bacteria to natural environments may drive resistance evolution by generating a pool of resistance mutations, where selection could act to enrich resistant mutants when antibiotic exposure occurs [470].

#### 6.4.3. Phylogenetic Limitations: Species Specificity of the Antibiotic Resistance Genes

Recent studies on transferring resistance-conferring mutations and full resistance genes into *E. coli*, from and to closely related species, have not been very successful. Resistance mutations originated from one bacterial species and transferred to another, and they rarely invoke a resistance phenotype. More frequently, the yield was drug hypersensitivity in close relatives, since the new gene could not fit the epistatic system of the new host, as was demonstrated in the case of the transferred aminoglycoside resistance (trkH) gene [471].

### 6.5. The CRISPR/Cas Bacterial Immune System: A Molecular Tool to Get Rid of Unwanted Antibiotic Resistance

Bacteria and Archaea have developed a system based on clustered, regularly interspaced, short palindromic repeats (CRISPR) from peculiar genetic loci. These repeats provide acquired immunity against viruses and plasmids by targeting the nucleic acid in a sequence-specific manner [472]. These hypervariable loci take up genetic material from invasive elements and build up inheritable DNA-encoded immunity over time. It was demonstrated that the *S. thermophilus* CRISPR1/Cas system can also naturally acquire spacers from a self-replicating plasmid containing an antibiotic-resistance gene, leading to plasmid loss. The most recent evolutionary classification of CRISPR-Cas systems and cas genes includes two classes, 6 types, and 33 subtypes. The discovery of numerous derived CRISPR-Cas variants is often associated with mobile genetic elements that lack the nucleases required for interference [473].

## 7. Clonally Evolving Pan-Genomic ESKAPE Pathogens

### 7.1. Pseudomonas aeruginosa: A Pan-Genomic Hotbed of Multidrug Resistance

The latest review on antibiotic resistance mechanisms in *P. aeruginosa* and alternative therapeutic strategies came out last year [474]. Our latest knowledge about the lifestyle of *P. aeruginosa* (for taxonomy: see Wikipedia) is that of a ubiquitous [475], invasive [476], opportunistic [477,478,479,480], facultative, pathogenic bacterium species. It can cause diseases in plant and animal species, as well as in humans [481], which has recently been summarized [482]. As a consequence of its evolutionary plasticity [483], based on open genome expandability [484,485,486,487], it is armored with a full arsenal of antibiotic multiresistance [3,488,489]. The majority of MDR *P. aeruginosa* displays a wide repertoire of antibiotic resistance mechanisms, including posttranslational modification of drug targets [490,491,492], enzymatic inactivation of antibiotic molecules [493], encoding genes of carbapenem-resistance preferably those coding for metallo-β-lactamase; overexpressing efflux pumps [494,495,496], and showing strong ability to form a biofilm [497,498,499].

*Pseudomonas aeruginosa* is one of the two pangenomic [499,500], Gram-negative pathogenic species, which has permanently acquired MDR encoding genes. It is a threatening source of resistance genes, which are transmittable via horizontal gene transfer [339,501].

#### 7.1.1. A Shortlist of Intrinsic and Acquired Antibiotic Resistance in *P. aeruginosa*

The most recent overviews [487,489] provides “an ocular perspective” of the different mechanisms of antibiotic resistance in *P. aeruginosa*. Most strains are intrinsically resistant to third-generation cephalosporins due to chromosomal-encoded C beta-lactamase and AmpC [488,501]. The original intrinsic MDR-arsenal includes the production of beta-lactamases, loss of outer membrane proteins [500], and up-regulation of efflux pumps [502,503,504,505,506,507]. This species also acquired resistance to aminoglycosides and fluoroquinolones [487]. One of the acquired enzymes taken-up by *P. aeruginosa*, *Pseudomonas* extended resistance (PER) [508,509,510,511], a class of extended-spectrum beta-lactamase (ESBL), occurs less frequently but still is of clinical importance [512,513].

*Pseudomonas aeruginosa* uses different mechanisms that can jointly contribute to its multiresistant phenotype [512] and multidrug efflux systems [492,495,496,514,515]. All of this makes *P. aeruginosa* extremely invasive. The rapidly increasing number of new *P. aeruginosa* isolates of MDR, XDR, and PDR phenotypes severely reduces the antibiotic therapy options available [3]. Valuation of the economic cost of antimicrobial resistance (AMR) is important for decision making and should be estimated accurately, [516]. Colistin, which acts on outer membranes of bacteria resulting in its permeability and cell-death, was suggested five years ago as a salvage therapy in the treatment of life-threatening infections due to MDR *P. aeruginosa* blood-stream infections (BSI) [517]. Articles and reviews on resistance problems appeared one year later [518] and have been appearing since [205,214,215,216,217].

#### 7.1.2. Pseudomonas Genetics and Genomics

Genetics: Most genes of intrinsic resistance are on the chromosome, including multidrug efflux pumps and enzymes responsible for resistance to beta-lactam and aminoglycoside antibiotics [519].

Genomics: A challenging option, provided by the next-generation whole-genome sequencing, as an approach to better understand and combat antibiotic resistomes [520], has been beneficially used in *Pseudomonas* research, as exemplified by Cao et al. [521]. An international consortium has been continuously providing comparative genomic information [522,523] and the prognosis [500] for clinical use. In addition. an antimicrobial resistance prediction concluded from the comparative genomics data of comparative sequence analysis of all available drug-resistant *P. aeruginosa* genomes could be realized [486,501,516,522,523].

Ramanathan and his associates studied single-nucleotide polymorphisms (SNPs) in ten multiantibiotic resistant *P. aeruginosa* clinical isolates [519]. Non-synonymous single-nucleotide polymorphisms (nsSNPs) were found in more clinical isolates compared to the reference genome (PAO1). The nsSNPs identified in the multidrug-resistant clinical isolates were found to alter a single amino acid in several antibiotic-resistant genes. They also found mutations in genes encoding efflux pump systems, cell wall, DNA replication, and genes involved in repair mechanisms. Furthermore, nucleotide deletions in the genome and mutations leading to a generation of stop codons were also observed in the antibiotic-resistant clinical isolates, and in specific mutations within antibiotic-resistant genes, compared to the susceptible strain of the same bacterial species [519].

#### 7.1.3. Transcriptomics

So far, we have detailed knowledge of the behavior of *P. aeruginosa* under standard laboratory conditions, but we only have a superficial understanding of bacterial functions and behaviors during human infection. However, a recent study on transcriptomes in human infection revealed that multiple genes known to confer antibiotic resistance had substantially higher expression in human infection than under laboratory conditions [524].

#### 7.1.4. Resistance Phenotypes

The first comprehensive review on this subject was published by Kempf and Rolain [525]. Mutations in pmrB confer cross-resistance between the LptD inhibitor POL7080 and colistin in *P. aeruginosa* [217]. Integrated whole-genome screening revealed that *P. aeruginosa* virulence genes use multiple disease models, and the pathogenicity is host-specific [8]. Biofilm formation is an important part of the pathogen strategy of this bacterium [526]. A list (incomplete) of *Pseudomonas*-caused human diseases is discussed in Appendix A.

### 7.2. Acinetobacter baumannii

#### 7.2.1. Species with a Gradually Expanding (Open) Pangenome

The bacterium *Micrococcus calcoaceticus* was isolated from soil by enrichment in a calcium-acetate-containing minimal medium [527]. It was finally moved to Moraxella [528] as a single representative of genus *Acinetobacter* in Bergey’s Manual [529] under the name of *A. calcoaceticus* (type strain ATCC 23055) [529]. *Acinetobacter baumannii* was designated taxonomically after appearing as a human pathogen [530,531,532,533]. *Acinetobacter baumannii* is also a veterinary pathogen [534,535,536,537,538], but, unlike *P. aeruginosa*, it has not appeared as a plant pathogen. *Acinetobacter baumannii* is a very successful pathogen [539]. WHO declared that *A. baumannii* is one of the two most serious pangenomic ESKAPE organisms [540]. Since then, the rapid global evolution of MDR in *A. baumannii* has been carefully monitored, and clonal lineages of old and new isolates have been revealed [541]. Two long-term international global surveys have recently been carried out on *A. baumannii* as a potential health risk factor, as reported by Gales et al. and Flamm et al. [540,541].

#### 7.2.2. Resistance Mechanisms and Diseases

The mechanisms of disease caused by *A. baumannii* strains have recently been reviewed by Morris et al. [8]. The latest review on the resistance mechanisms of *A. baumannii* [542] focuses on XDR resistance.

As for its resistance mechanisms, tigecycline efflux was described as a mechanism for non-susceptibility in *A. baumannii.* In addition, a deletion of TnAbaR23 resulted in significant antibiogram changes in a multidrug-resistant *A. baumannii* strain [543]. For more information see Appendix A.

#### 7.2.3. Genetic Dissection of Colistin Resistance in *A. baumannii*: Trebosc et al., 2019

The transcriptional regulator PmrA was considered as the only potential drug target to restore colistin efficacy in *A. baumannii* [510], but the deletion of *pmr*A, the gene responsible for overexpression of the phosphoethanolamine, PetN, only restored susceptibility in a few strains. A detailed genetic analysis revealed a new colistin resistance mechanism mediated by genomic integration of the ISAbaI insertion element upstream of the PmrC homolog (*ept*A), leading to its overexpression. The gene *ept*A is present in each international clone 2 clinical strain, and a duplicated ISAbaI-eptA cassette was present in at least one clinical isolate, thus indicating this colistin resistance determinant may be embedded in a mobile genetic element [216].

#### 7.2.4. Genetics Toolkits to Becoming Multiresistant

*Acinetobacter baumannii*, like other non-glucose-fermenting Gram-negative species such as *P. aeruginosa*, has increasingly been acquiring carbapenem resistance [544,545]. An argument against the worries [546] that *A. baumannii* increases the risk of carbapenemase spread in general, since horizontal resistance gene the transfer can occur between Gram-negative species, regardless of their ability to ferment glucose [545], is that experimentally transferred resistance mutations from other species to *E. coli* have been mostly silent, and frequently yield drug hypersensitivity [469].

#### 7.2.5. Sword of Damocles: Clonal Evolution, Global Spread, and Epidemic Potential of *A. baumannii*

The different antibiotic resistance genes are organized into resistance islands in the genome of *A. baumannii*. The impression of the reviewers is that the MDR, XDR, and PDR must not be considered as simple collections of different antibiotic resistances, but they may represent a coordinated system based on sophisticated genetic background and genomic organizations. Based on the evolutionary analysis of six housekeeping genes, *A. baumannii* is of monophyletic origin [531,546]. Originally three predominant pathogen clones, called ‘international clonal lineages’ (ICLs), were known as being responsible for hospital outbreaks worldwide [547]. The monophyletic status of ICLs 1 and 2 have also been shown [38,548]. As reviewed first by Dijkshoorn et al. [546], the genome of a representative ICL1 (AYE) [549,550,551] strains include 52 genes associated with resistance to anti-microbial drugs, and 45 are localized in an 86 kb resistance island called AbaR5 [552]. ABAR1 is also present in other *A. baumannii* strains, but at a much smaller size. The presence of an extraordinary 22 gene-cassette coding for transposases and insertion sequences may be responsible for the acquisition of resistance genes into the AbaR1 island of the AYE ICL1-type strain [543]. Almost half are orthologous to coding sequences of *Pseudomonas* [546].

#### 7.2.6. Genome Plasticity

The *Acinetobacter baumannii calcoaceticus* complex is considered as a species with an open pan-genome and is well-defined on Wikipedia. The complete genome [553,554,555] includes the core genome (genes present in all isolates) with extreme antibiotic resistance traits [554]. In addition, the accessory genome [555], including the genes absent from one or more isolates, or unique to a given isolate, also contains hosts antibiotic resistance genes but in different arrangements [555]. Antibiotic resistance genes are located both in the core and the accessory genomes [556]. In the latter, they were found in alien islands or flanked by integrases, transposases, or insertion sequences [556]. They must be acquired via horizontal gene transfer from other *Acinetobacter* strains to colonize the same environment [556]. The first detailed genomic analyses revealed that the existing *A. baumannii* clinical population consists of low-grade pathogens, whose pathogenicity relies mainly on an ability to persist in the hospital setting and survive antibiotic treatment [4]. This analysis led to the conclusion that *A. baumannii* has a high capacity to acquire new genetic determinants and displays an open pan-genome. This feature may have played a crucial role in the evolution of this human opportunistic pathogen towards clinical success [4]. The whole pangenome of *A. baumannii* consists of >8800 orthologous coding sequences, and it has exponentially been increasing as new genomes become available (an open pan-genome), mainly due to unique accessory genomes of different isolates enriched with acquired genes of transport and transcription regulation functions [4]. Since then, at least 15 complete, and 180 draft, chromosomal *A. baumannii* genomes, 31 plasmids, and six bacteriophage sequences have been available on the NCBI database, together with those of other species of the genus, and antibiotic treatment induces important genes [557]. The open pan-genome of *A. baumannii* also includes lots of plasmids, transposons, integrons, and genomic islands, which may contribute to the evolutionary success of this clinical pathogen. This is particularly true in the acquisition of multidrug resistance determinants [5], a tool for redefining this, and other, pathogenic bacterium species [162,547,557,558].

#### 7.2.7. Will the Epidemic Threat Be Materialized?

The population structure of *A. baumannii* comprises of a set of expanding multiresistant clones raised from an ancestral susceptible genetic pool [559,560,561], confirming that *A. baumannii* used to have low phylogenetic diversity, providing a narrow evolutionary bottleneck through which a micro-evolutionary tree with many branches has emerged [546]. Making this metaphor complete, each branch of that tree carries “dangerous fruit” called antibiotic-resistance (ABR) genes. All the other known *Acinetobacter* species are harmless soil-inhabiting ones [546].

Until recent years, *A. baumannii* has been almost exclusively isolated from locations of antibiotic stress conditions, that is, in hospital environments, (especially from intensive care units of hospitals) and veterinary clinics [562,563,564]. However, in 2017, *A. baumannii* DSM30011 was obtained from the resinous desert shrub guayule and may be considered as a “copy” of the first isolate, and whole-genome sequencing and phylogenetic analysis based on core genes confirmed DSM30011 affiliation to A. baumannii [565]. No antimicrobial resistance islands were identified in DSM30011, agreeing with a general antimicrobial susceptibility phenotype. The marginal ampicillin resistance of DSM30011 most likely derived from chromosomal ADC-type *amp*C and blaOXA-51-type genes [566]. The environmental *Acinetobacter baumannii* isolate DSM30011 reveals clues into the preantibiotic era genome diversity, virulence potential, and niche range of a predominant nosocomial pathogen [567] armed with efficient multidrug efflux pumps [567].

The “branches” of the original “tall” “micro evolutionary tree” are rather “long.” ln the frame of a SENTRY study [543], *Acinetobacter* clinical isolates from 10 South Asian and Pacific countries were tested for being non-susceptible to imipenem or meropenem and positive for OXA-23-, OXA-24/40-, OXA-58-, and MBL-encoding genes [540,541]. The sword of Damocles (a global *Acinetobacter* epidemic) is still possible, but has not yet been dropped down.

## 8. Concluding Remarks

We must apologize for omitting so many excellent publications from this field because of space limits.

### 8.1. Forcasts Based On Evolutionary Data

#### 8.1.1. The Main Question

Whether the post-antibiotic era is at the gate, or there is a chance to prevent its coming, is the question of questions. The literature convinced us that the trend of the global expansion of clonally spreading MDR and XDR pathogens is not irreversible. The post-antibiotic era is not inevitable. Let us discuss the arguments for and against these statements.

#### 8.1.2. The “Combat Scenario”

Nature and science have long been battling. Science produces antibiotics, nature presents pathogens with resistances. Nature has gradually become more sophisticated and better organized. The MDR or XDR phenotype cannot be considered as a brute summary of the expression of ad hoc collected (intrinsic and acquired) antibiotic resistance alleles. MDR (or XDR) represents a concerted phenotypic expression of structurally and functionally organized (operons, cassettes, integrons), and coordinated (resistomes) resistance alleles, which had been selected by the evolution. Both the intrinsic occurrence and the uptake of resistance alleles are random.

#### 8.1.3. Adaptive Evolution

The selective drives acting in the commensal niche and those acting in a milieu under selective (antibiotic) pressure are different. A neutral mutant allele, including those encoding resistance, in a commensal niche may be fixed in the population on the condition that it does not display a heavy genetic load, that is it does not express a negative fitness affecting (DFE) phenotype]. That is, it does not have either a positive or a negative selective value. Therefore, new antibiotic resistances are expected to appear in the future as in the past.

Neutral mutant alleles, including resistance-encoding ones, in a commensal niche may be fixed in the population on the condition that it does not display a heavy genetic load, or if it does not express negative fitness affecting (DFE) phenotype. That is, it does not have either a positive or negative a selective value. Therefore, new antibiotic resistances are expected to appear in the future, as in the past.

A mutant allele with a strong antibiotic resistance phenotype will expectedly be lost, or at least prevented from domination, or selectively lost in commensal conditions if it represents a heavy genetic load. This phenomenon is demonstrated in the monitoring studies on Gram-negative and Gram-positive MDR pathogens. However, the same allele will be enriched in the population up to 100% under selective (antibiotic) stress conditions, either in a hospital environment or in the laboratory.

#### 8.1.4. The “Dialectics” of Resistance and Sensitivity

There are exceptional situations when the same gene can act as a resistance or a sensitivity gene, in the same species, depending upon genetic background (Appendix A).

#### 8.1.5. Collateral Sensitivity: A Biochemically Proven Limiting Factor in MDR Evolution

For evolutionary success (winning the “permanent war”), the pathogen needs to carry strong and heavy “weapons” (MDR combinations), which also increases the genetic load. The rediscovery, genomic, and evolutionary interpretation of the collateral sensitivity is the most important experimental proof that the resistance capacity of a bacterium is limited. It means that “trees cannot grow up to the sky”. The credibility of this discovery is also supported by biochemical evidence]. The molecular technique for getting rid of an unwanted resistance is probably the CRISPR/Cas Bacterial Immune System.

This conception is illustrated in an allegoric way on Figure 2. The higher resistance level makes the pathogen stronger in selective but weaker in non-selective (commensal) conditions. (Figure 2A–H). This explains collateral sensitivity as well. A limited number of resistances can be harbored, but not more. If the pathogen gains a new one (#11 in Figure 2G, and #13 in Figure 2H) it must get rid of another one (#1 Figure 2G and #3 on Figure 2H). The data in the last report of the HAI Consortium on changes in antibiotic resistance profiles of MRSA clones in the USA indirectly confirm that collateral sensitivity is not a laboratory artifact [256].

#### 8.1.6. Counterarguments

An experimentally supported counter-argument is compensatory evolution [461]. Another fact taken into considerations is that neither *E. coli* nor *S. aureus* is a pangenomic species. It is unknown whether the pangenomic pathogen species (*A. baumannii* [532], *P. aeruginosa* [462], *K. pneumoniae* [225]) were able to compensate for the genetic load by expanding their genome. They can withstand an increased number of resistances by exploiting their evolutionary plasticity and disseminate clonally or polyclonally. They serve as a permanent hotbed of other MDR pathogens [439]. Fortunately, there are phylogenetic barriers to horizontal transfer of antimicrobial peptide resistance genes in the human gut microbiota [466,469]. Another optimistic message from experimental evolution is that the majority of antibiotic resistance genes are species-specific and do not increase, but rather decrease, the MDR phenotype if they are uptaken by *E. coli*. [469].

Theoretically, however, epidemics caused by any of the three pangenomic pathogenic species might break out at any time. Our view is that since all but *K. pneumoniae* should be considered as a first-grade pathogen, the epidemic danger may not be immediate. However, immunocompromised patients in hospital environments are at permanent risk.

### 8.2. Statistic-Based Conclusions from the Frequencies of MDR-Related Publications in PubMed

We drew some conclusions from the temporal distribution of MDR literature in the last decade. The publications convinced us that only the monoclonal-spreading pangenomic pathogen species display a global epidemic threat. Naturally, it should not be underestimated. However, the nearly even temporal distribution of the literature seems to be an indicator of a permanently dramatic, rather than an alarming, situation.

All three tables should be evaluated together. Table 1A–C shows a dramatic increase in the number of reports on new resistant isolates of each Gram-negative ESKAPE species. No report was found about XDR or PDR, concerning the non-pan-genomic “ESKAPE Club members” (see also Table 2). When we entered MDR/XDR, or pan-resistant, and *S. aureus* or *Enterococcus*, we invoked few (24 and 6, respectively) references. However, none of those strains discussed in the articles would qualify as pan-resistant or XDR. We propose that the plausible explanation must be collateral sensitivity. The polyclonally spreading pathogens also display a permanent threat but would not cause epidemics. As far as we know, there is no clonally disseminating pan-genomic Gram-positive pathogen species. In addition, we found few references about Gram-positive XDR isolates, although <5% of the publications about PDR isolates are cited by PubMed. However, the evidence on pan-genomic isolates contradicts this conclusion. We suppose that there are other unrevealed resistance mechanisms based in the versatility of Gram-positive cell walls, which may provide more general protection to the Gram-positive organisms expressing the respective alleles, but we do not suppose that this kind of resistance could be disseminated globally.

Table 3 demonstrates that literature of the two pangenomic Gram-negatives is the largest, and many of the publications deal with MDR, XDR, and PDR problems. The only significant difference between the two species is that *Pseudomonas* has a large host range, including plants, while the host range of *Acinetobacter* is extremely narrow. We did not focus on *Klebsiella* in this review. Very few of the *Acinetobacter* and *Pseudomonas* publications deal with collateral sensitivity, and that is not good news. It probably indicates that this evolutionary mechanism is not working in pangenomic bacteria. It should urgently be experimentally tested.

### 8.3. Pharmaceutical Perspectives

#### 8.3.1. Search for Omnipotent (“Jolly Joker”) Antibiotics

We feel the clinching argument for our view of science is as follows. So far, science has produced “cards”, including the ace cards” (colistin, daptomycin) for the “card games”, but each was “hit” by the respective resistances. “Jolly Jokers” are needed. One “Jolly Joker” (teixobactin) [383] active in Gram-positive bacteria has been identified. So far, no resistance has been noted [388].

#### 8.3.2. The Perspectives of Antimicrobial Peptide (AMP) Molecules

The different types of multidrug resistances (MDR, XDR, PDR) are an enormous challenge with clinical, veterinary, and plant-pathogenic significance. This review aimed to give a general view rather than specific perspectives. We are aware that chemotherapeutic tools should always be available, ready to combat new pathogens and overcome any new resistance. We are convinced that due to their molecular heterogeneity, structural versatility, and target spectra, the natural antimicrobial peptide (AMP) molecules and synthetic derivatives provide an inexhaustible source of new optimized toolkits capable of combatting new MDR pathogens. Furthermore (as discussed in 6.4.1), both the mobilization patterns, as well as the functional compatibilities with new target bacterial hosts of AMP-resistance genes, and traditional antibiotic-resistance genes, are different.

The AMP research strategy relies on searching for new natural molecules, computer-design and quantitative structure–activity relationships (QSAR), and analysis of derivatives to provide databases for getting descriptor molecules with better pharmacokinetic profiles. Many natural AMPs are primary (ribosome-templated) gene products. Others are or non-ribosomal-templated peptide (NRP) molecules, synthesized enzymatically by multienzyme thiotemplate mechanisms using non-ribosomal peptide synthetases (NRPS) and/or fatty acid synthase (FAS)-related polyketide synthases. Our next review (Fodor et al., in preparation) will provide a useable inventory of AMPs. This includes primary gene products (from mammals, free-living bacteria, plants, plant-bacterium symbiotic associations, and insects) and NRPs. In that review, we intend to discuss the role of natural AMPs in plant innate immune defense mechanisms, especially genetic aspects discovered in *Arabidopsis thaliana*.

The specificity of the resistance mechanism is based on Microbe-Associated Molecular Patterns (MAMPs). The anti-MDR potential of the large target-spectral NRP-AMPs produced by obligate bacterial symbiotic bacteria (EPB, belonging to the genera *Xenorhabdus, Photorhabdus*) of entomopathogenic nematodes (EPN, *Steinernema, Heterorhabditis* species) in vitro is excellent. Their natural role is to provide monoxenic conditions for the respective EPN/EPB symbiotic “couple” in polyxenic (soil, cadaver) environments. They are effective against a large number of both prokaryotic and eukaryotic opportunistic soil microbes, as well as against human, veterinary, and plant-pathogen organisms which they have never encountered in nature (as noted in several labs). They might successfully be applied more efficiently for plant protection as soil ingredients than in clinical practice.

As discussed in Section 4.2.6, Section 4.2.7 and Section 5.1, some NPR-AMPs (vancomycin, daptomycin, teixobactin) were efficient against Gram-positive targets. All but teixobactin has been overcome by resistance. Others (colistin, Section 3.6 and Section 3.7) were used against Gram-negative MDR targets, but have also been overcome by resistance. Some drug candidates with unnatural (beta) amino acid substituents are extremely efficient against ESKAPE pathogens [383,391]. Others (ARV-1502) are in clinical studies [402,403].

As discussed in Section 5.2, the proline-rich AMPs (PrAMPs of insect origin and their optimized derivatives) are of a uniquely outstanding medical perspective, especially because they have a larger target spectrum in vivo than in vitro. These data confirm our previous conclusion that the battle between newly appearing representatives of MDR and novel AMPs is permanent and endless.

We note the availability of recently available AMP databases that could be used for modeling. The best ones are as follows: DBAASP (PubMed: 27060142), APD3 (PubMed: 26602694), CAMP (PubMed: 19923233), LAMP (PubMed: 23825543), and dbAMP (PubMed: 30380085), just to mention a few. In addition, Spanig and Heider recently published a review on AMP modeling for diagnostics (PubMed: 30867681), which we referred to as [391].

### 8.4. Closing Remark

Apart from natural reasons (phylogenetic barriers, collateral sensitivity), the profession ability of molecular designers and the availability of abundant natural sources of antimicrobial peptides with structural diversity and heterogeneity can assure the post-antibiotic age will never come. The question is whether nature will, or will not, be able to invent new wedges (resistance) against the new potential “Jolly Jokers” (AMPs).

## Figures and Tables

**Figure 1 pathogens-09-00522-f001:**
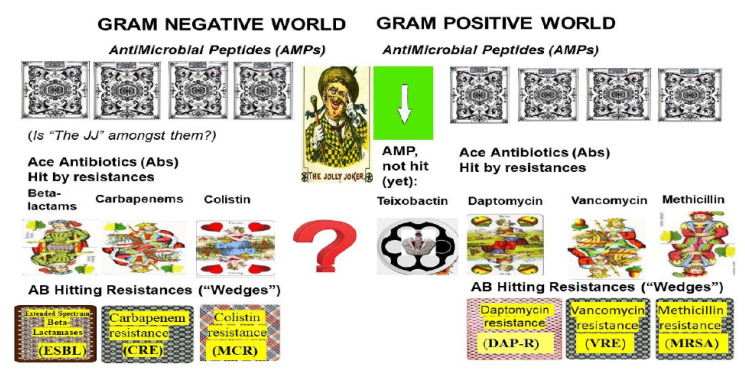
**Illustration of the competition between new antbiotics and invoed resistances.** The “Card Game” between science (designers of antibiotics) and nature (antibiotic resistance profile designing pathogens). Legend to Figure 1. Both with the Gram-negative (**left**) and the Gram positive (**right**) “card tables”, the respective uppermost row represents the cards in the hands of science (antibiotics, antimicrobial peptides), and the lowest line is the cards (resistances) in the hands of nature. Science put down the “first card”, penicillin (let us refer to it as a “Jack”). However, nature replied with the “wedge”, called penicillin-resistance, acting as a trump-card to hit “Jack” in both the Gram-negative and the Gram-positive “card games”. Then, science put down its’ “queen” to hit this “wedge” in both “card-games”: the beta-lactams, such as the amino-penicillin family for the Gram-negative and methicillin for Gram-positive “card games”. They worked properly until nature produced new “wedges” (resistances): the extended spectral beta-lactamases (ESBL) and methicillin-resistance (MRSA), acting as trump cards in the Gram-negative and the Gram-positive “card games”, respectively. Then, science put down “kings” to hit the queen-hitting wedges: carbapenems to overrule ESBL, and vancomycin to overrule MRSA in Gram-negative and Gram-positive “card games”, respectively. Soon after the introduction of the “kings”, nature produced king-hitting trump card “wedges” such as CRE (carbapenem resistance) in the Gram-negatives and vancomycin-resistance (VRE in Enterococci and VRS in Staphs) in the Gram-positive “card games”. “Ace” antibiotics have become urgently needed. The Science took out the “old card” colistin (polymyxin) as an “ace” against the Gram-negative, and the newly discovered antimicrobial peptide against Gram-positives. However, Mother Nature produced new “wedges” again, the colistin- and daptomycin- resistance bacteria. As the game ramps up, “Jolly Jokers” are now needed and being searched for. The first potential “Jolly Joker”, teixobactin [382], was active on Gram-positives. It was isolated and identified 4 years ago, and since then has not invoked resistance. Additional “Jolly Joker” antibiotics acting against both Gram-positive and Gram-negative targets are still needed.

**Figure 2 pathogens-09-00522-f002:**
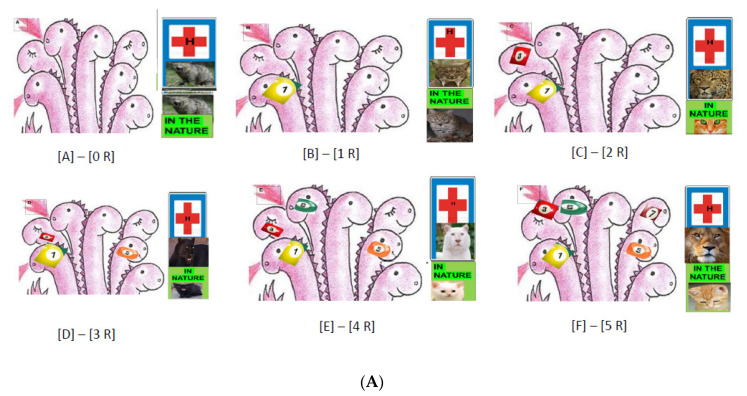
**Illustration of Collateral Sensitivity.** Trees do not grow to the sky: a metaphoric illustration that the MDR pathogens cannot be overwhelming winners in nature. In Figure 2, the pathogen bacterium is illustrated as a seven-headed monster. (**A**) demonstration of different trends of adaptive evolution in commensal and hospital environments. (**A**) Each new antibiotic resistance ([A]—[0 R]; [B]—[1 R];[C]—[2 R]; [D]—[3 R]; [E]—[4 R]; [A]—[5 R]) elevates the genetic load and reduces fitness, which makes the pathogen more vulnerable in antibiotic-free (commensal) conditions symbolized as nature (**A**). Under antibiotic stress conditions, hospital resistance as a positive selection marker makes the pathogen even more powerful and dangerous. (**A**) A pathogen without antibiotic resistance (R = 0) has similar strength in the hospital and in nature (the pathogen is symbolized as a wild cat). (**B**) One antibiotic resistance (R = 1) (#1) makes the pathogen a little stronger (an ounce) in the hospital and a little weaker (bobcat) in nature. (**E**,**F**) The more resistant alleles (#1, #3, #5, #6 and #1, #3, #5, #6. #7) that are present, the more strength in the hospital (symbolized ounce and lion, respectively) and elevated weakness (“baby cats”) in nature. (**B**) Demonstration of collateral sensitivity. (**G**,**H**) The load-bearing potential must be limited, at least the phenomenon of collateral sensitivity seems to support this forecast. When the hypothetical seven-fold resistant pathogen (resistant to antibiotics #1, #3, #5 #6, #7, #8, and #9) acquires an 8th (#11*, (**G**)) and a 9th (#13*, (**G**,**H**)), resistance, respectively, it drops out resistance #1 and #3, respectively.

**Table pathogens-09-00522-t001A:** **A.** Temporal Distribution of PubMed Cited Reviews on ESBL, Carbapenem, and Colistin Resistance in Gram-negative (Old List) ESKAPE bacteria

Questions to PubMed (Entries)	Answers (Items): Articles: A, Reviews: R)					
	ALL		2019	2018	2009	2001
	A	R	R	R	R	R
Beta-lactams	130,408	9814	121	325	345	332
Extended Spectrum Beta-Lactam Resistance (ESBL)	5906	510	46	64	20	11
ESBL *Klebsiella*	2763	192	11	13	10	8
ESBL *Escherichia coli*	3952	191	5	20	12	6
ESBL *Acinetobacter*	505	64	5	4	6	1
ESBL *Pseudomonas*	853	98	7	8	6	0
Carbapenems	17,031	1992	135	193	114	34
Carbapenem Resistance (CRE)	11,981	1461	118	164	95	20
CRE Enterobacteriaceae	2085	344	50	64	7	0
CRE *Klebsiella*	3067	292	22	42	19	4
CRE *Acinetobacter*	2959	334	36	32	23	3
CRE *Pseudomonas*	3142	377	28	36	22	5
Colistin	7027	653	72	78	21	4
Colistin resistance	4640	509	62	63	18	3
*mcr* colistin resistance	625	39	7	13	0	0
Colistin resistance, Enterobacter	1712	150	20	29	2	0
Colistin resistance, *Klebsiella*	914	111	12	13	3	0
Colistin resistance, *Acinetobacter*	1363	185	22	17	9	1
Colistin resistance, *Pseudomonas*	1123	154	10	14	11	2

**Table pathogens-09-00522-t001B:** **B.** Temporal Distribution of PubMed Cited Reviews on Multidrug- Resistance (MDR), Extended Spectrum (XDR) Resistance, and Pan-resistance in Clonal and Polyclonal Disseminating Gram-Negative ESKAPE Pathogens

Questions to PubMed (Entries)	Answers (Items): Articles: A, Reviews: R)					
	ALL		2019	2018	2009	2001
	A	R	R	R	R	R
*Acinetobacter* MDR	1261	174	25	25	10	0
*Acinetobacter* MDR/XDR	18	2	0	0	0	0
*Acinetobacter* Pandrug-resistant	96	23	2	4	2	0
*Pseudomonas* MDR	1310	165	19	27	9	2
*Pseudomonas* MDR/XDR	25	4	0	1	0	0
*Pseudomonas* Pandrug-resistant	52	1	1	2	1	0
*Klebsiella* MDR	1056	100	18	14	4	1
*Klebsiella* MDR/XDR	9	2	1	1	0	0
*Klebsiella* Pandrug-resistant	48	11	1	1	0	0
*Enterobacter* MDR	272	14	5	1	1	0
*Enterobacter* MDR/XDR	8	2	1	0	0	0
*Enterobacter* Pandrug-resistant	1	0	0	0	0	0
*cephalosporin resistant Enterobacteriaceae* MDR	121	12	2	3	0	0
*cephalosporin resistant Enterobacteriaceae* MDR/XDR	1	0	0	0	0	0
*cephalosporin-resistant Enterobacteriaceae* Pandrug-resistant	0	0	0	0	0	0
*Salmonella typhi* MDR	177	23	2	3	1	0
*Salmonella typhi* MDR/XDR	8	1	0	0	1	0
*Salmonella typhi* Pandrug-resistant	0	0	0	0	0	0
*Escherichia coli* MDR	1721	107	12	12	6	3
*Escherichia coli* MDR/XDR	8	1	0	0	0	0
*Escherichia coli* Pandrug-resistant	24	3	0	0	0	0

**Table pathogens-09-00522-t001C:** **C.** Temporal Distribution of PubMed Cited Reviews on Multidrug- Resistance (MDR), Extended Spectrum (XDR) Resistance, and Pan-resistance in Bacteria Recently Scored to the Gram-Negative ESKAPE Pathogens [119]

Questions to PubMed (Entries)	Evoked Items: Articles: A, Reviews: R)					
	ALL		2019	2018	2009	2001
	A	R	R	R	R	R
*Helicobacter pylori* MDR	54	2	0	1	0	0
*Helicobacter pylori* MDR/XDR	0	0	0	0	0	0
*Helicobacter pylori* Pandrug-resistant	1	1	0	0	0	0
clarithromycin-resistant *Helicobacter pylori*	316	33	3	5	0	1
clarithromycin-resistant *Helicobacter pylori* MDR	4	0	0	0	0	0
clarithromycin-resistant *Helicobacter pylori* MDR/XDR	0	0	0	0	0	0
clarithromycin-resistant *Helicobacter pylori* Pandrug-resistant	0	0	0	0	0	0
*Campylobacter* MDR	72	5	0	1	1	0
*Campylobacter* MDR/XDR	0	0	0	0	0	0
*Campylobacter* Pandrug-resistant	0	0	0	0	0	0
fluoroquinolone-resistant *Campylobacter*	69	10	4	3	2	3
fluoroquinolone-resistant *Campylobacter* MDR	4	0	0	0	0	0
fluoroquinolone-resistant *Campylobacter* MDR/XDR	0	0	0	0	0	0
fluoroquinolone-resistant *Campylobacter* Pandrug-resistant	0	0	0	0	0	0
*Neisseria gonorrhoeae* MDR	44	3	0	0	0	0
*Neisseria gonorrhoeae* MDR/XDR	0	0	0	0	0	0
*Neisseria gonorrhoeae* Pandrug-resistant	0	0	0	0	0	0

**Legend to Table 1**: The number of publications on anti-Gram-negative compounds in beta-lactams is an order of magnitude larger than those publications on carbapenem. The number of publications about CRE is about a half order of magnitude larger than about ESBL and gradually increasing, especially in Enterobacteriaceae. The majority of colistin resistance-related publications appeared in the last decade, but each resistance seems to be a rather permanent, rather than alarming problem, if we consider the number of reviews as an indicator (Table 1A). The publications found by entry on MDR are very high for each ESKAPE Gram-negative pathogen, with *Escherichia coli* showing a significant but not sharp increase. However, MDR/XDR evoked a significant number of items, only on the clonally distributing ESKAPE pathogens, and few of the others, including *E. coli*. The entry PDR evoked a significant number of items, only on the clonally distributing ESKAPE pathogens, and few of the others, except for *E. coli* with 24 items (Table 1B). As for the new “ESKAPE Club members”, only one publication reported PDR (in *Helicobacter*), and none for XDR. However, the entry MDR invoked a significant number of recent publications (Table 1C).

**Table 2 pathogens-09-00522-t002:** Temporal distribution of PubMed cited reviews on MDR-related issues in Gram-positive ESKAPE bacteria.

Questions to PubMed (Entries)	Evoked Items: Articles: A, Reviews: R)					
	ALL		2019	2018	2009	2001
	A	R	R	R	R	R
**A.** Temporal Distribution of PubMed Cited Reviews on MDR-related issues with MRSA *Staphylococcus aureus*						
*S. aureus* MRSA	31,827	2098	141	249	384	161
*S. aureus MRSA Antibiotic Resistance*	15,822	2085	35	83	148	109
*S. aureus Vancomycin* Resistance	6448	1119	22	36	73	60
*S. aureus MRSA VAN-R*	4591	885	13	28	50	42
*S. aureus Daptomycin* resistance	1462	319	11	17	26	1
*S. aureus MRSA DAP-R*	1169	264	10	13	18	2
*S. aureus* Carbapenem Resistance	1189	187	16	19	9	3
Teixobactin	69	13	1	4	0	0
Teixobactin, MRSA	15	3	0	0	0	0
*S. aureus* MDR	1122	119	9	17	9	2
MDR/XDR Gram-positive	21	5	1	2	0	0
MDR/XDR *S. aureus*	3	2	1	1	0	0
**B.** Temporal Distribution of PubMed Cited Reviews on MDR-related Issues in the genus *Enterococcus*						
*Enterococcus* Antibiotic Resistance	9167	1060	26	50	51	51
*Enterococcus* MDR	418	44	6	5	0	2
*Enterococcus* MDR/XDR	2	2	1	1	0	0
*Enterococcus methicillin-resistant*	2253	453	18	22	23	19
*Enterococcus* Vancomycin Resistance	5355	713	55	77	53	58
*Enterococcus* VRE	2247	274	10	20	6	14
*E. faecium* Vancomycin Resistance	2635	226	10	12	8	14
*E. faecium* VRE	985	83	5	6	2	7
*E. faecalis* Vancomycin Resistance	1824	241	6	11	6	4
*E. faecalis* VRE	564	45	1	4	1	3
*E. caecum* Vancomycin Resistance	13	0	0	0	0	0
*E. gallinarum* Vancomycin Resistance	278	10	11	2	6	7
*Enterococcus* Daptomycin resistance	584	120	30	42	35	7
*E. faecium* Daptomycin resistance	347	47	3	2	4	4
*E. faecalis* Daptomycin resistance	295	33	1	2	2	0
*E. caecum* Daptomycin Resistance	2	0	0	0	0	0
*E. gallinarum* Daptomycin Resistance	11	1	0	0	0	0
*Enterococcus* carbapenem resistance	524	72	7	5	2	2
**C.** Antibiotic Multiresistance (MDR), Extreme Large Spectrum (XDR) Resistance, and Pan-resistance in Gram-Positive ESKAPE Pathogens						
Gram-positive bacteria MDR	4292	542	11	42	34	12
Gram-positive bacteria MDR/XDR	21	5	1	2	0	0
Gram-positive bacteria PDR	17	6	0	1	0	0
*S. aureus* MDR	1122	119	9	17	9	2
*S. aureus* MDR/XDR	3	2	1	1	0	0
*S. aureus* Pandrug-resistant	19	9	0	2	0	0
*S. aureus* MRSA MDR	715	84	6	14	8	1
*S. aureus* MRSA MDR/XDR	2	2	1	3	0	0
*S. aureus MRSA* Pandrug-resistant	12	5	0	1	0	0
*Enterococcus* MDR	418	44	6	4	0	2
*Enterococcus* MDR/XDR *Enterococcus*	3	1	1	0	0	0
*Enterococcus* Pandrug-resistant	9	4	0	1	0	0
*Enterococcus* VRE MDR	92	12	3	4	0	0
*Enterococcus* VRE MDR/XDR	1	1	1	0	0	0
*Enterococcus* VRE Pandrug-resistant	3	0	0	0	0	0

**Legend to Table 2**: The bulk of the MRSA literature deals with antibiotic resistance. Interestingly, the temporal distributions of VAN-R and DAP-R MRSA publications are almost even, indicating a serious and permanent, but not an alarming, situation. A significant amount of the literature deals with carbapenem resistance in every year, and surprisingly few are about teixobactin in MRSA, as if this antibiotic, which was discovered 4 years ago, would not have been the dreamed for Jolly Joker antibiotic (Table 2A). Like polyclonally disseminating Gram-negative ESKAPE pathogens, despite the large number (1122) of MDR-subjected publications, only 3 items were found with MDR/XDR in a MRSA entry (Table 2C). The trend in *Enterococci* is rather similar. However, unlike the MRSA literature, the number of MDR articles is dramatically increasing for the *Enterococcus* literature, especially *E. faecium*, which seems to be in antibiotic-resistance evolutionary bloom. As far as we know, there is no known clonally disseminating pan-genomic Gram-positive pathogen species, and few references about Gram-positive XDR isolates were found, even though <5% of the publications about PDR isolates are cited by PubMed.

**Table 3 pathogens-09-00522-t003:** Temporal distribution of antibiotic resistance related PubMed cited reviews on genetics and genomics of the open pan-genomic Gram-negative pathogens *Acinetobacter baumannii* and *Pseudomonas aeruginosa*.

Questions to PubMed (Entries)	Evoked Items: Articles: A, Reviews: R					
	ALL		2019	2018	2009	2001
	A	R	R	R	R	R
***Acinetobacter***						
Intrinsic resistance	243	46	2	3	2	0
Acquired Resistance	1208	243	9	14	16	6
Resistance genes	1808	103	11	10	3	1
Plasmids, Genomic Islands	27	4	0	1	0	0
Antibiotic Resistance mechanisms	932	206	14	13	11	2
Antibiotic Resistance, Genome plasticity	24	4	0	0	0	0
Horizontal Gene Transfer	210	16	0	1	1	0
Pangenome pangenomic	9	1	0	0	0	0
Clonal Global Distribution	16	1	0	0	0	0
Diseases	4237	530	45	50	36	18
Host range	73	9	1	0	0	0
Collateral Sensitivity	5	1	0	0	0	0
***Pseudomonas***						
Intrinsic resistance	577	117	8	11	3	4
Acquired Resistance	2249	437	25	32	27	12
Resistance genes	4647	233	12	17	4	6
Plasmids, Genomic Islands	39	2	0	0	0	0
Antibiotic Resistance mechanisms	2075	411	16	35	16	10
Resistance, Genome plasticity	36	4	1	0	0	0
Horizontal Gene transfer	512	44	2	3	2	1
Pangenome	50	4	0	0	0	0
Clonal Global Distribution	6	0	0	0	0	0
Diseases	22.950	2628	92	157	129	04
Host range	404	48	2	8	4	1
Collateral Sensitivity	10	1	0	0	0	0

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
