# Peer review of "Multidrug Resistance (MDR) and Collateral Sensitivity in Bacteria, with Special Attention to Genetic and Evolutionary Aspects and to the Perspectives of Antimicrobial Peptides—A Review"

_pathogens, 2020, doi:10.3390/pathogens9070522_

Round 1
Reviewer 1 Report
The paper is well written and gives a nice overview of the current research in the field. However, I miss some information about available AMP databases that could be used for modeling. There are plenty of these databases out there, e.g., DBAASP (pubmed:27060142), APD3 (pubmed:26602694), CAMP (pubmed:19923233), LAMP (pubmed:23825543), dbAMP (pubmed:30380085), just to mention a few. Moreover, Spanig and Heider (pubmed:30867681) recently published a review on AMP modeling for diagnostics, which should be also discussed, at least in the concluding remarks.
Reviewer 2 Report
This article is poorly written with numerous grammatical mistakes, sentence structure issues, incomplete sentences and incorrect use of punctuation, that it was very hard to read it. They also did not italicize the names of bacteria consistently in the paper. This paper needs a very extensive revision before its science could be judged properly.
Round 2
Reviewer 1 Report
The authors addressed my concerns adequately.
Author Response
..

Reviewer 2 Report
This is a very comprehensive review article on multiple drug resistance. It is too long and may lose the interest of the audience. There are still a number of typographical and grammatical errors in the manuscript that need to be corrected before it is published. Some of them have been highlighted in the attached document. Even though it says it focuses on the antimicrobial peptides, but it is seems to be a minor part of the manuscript.
Author Response
I try, but the System does not allow me to do so.
Please let I allow tom upload my Response to Reviewr2.
The System refuses my repeated attempts.
András Fodor

Round 3
Reviewer 2 Report
I did not see the figures in any document.
Author Response
I would like to send the Figure material and the Graphic Abstract please let me do that.
